# Protective effects of klotho on palmitate-induced podocyte injury in diabetic nephropathy

Jeong Suk Kang[1,2], Seung Seob Son[1], Ji-Hye Lee[3], Seong Woo Lee[1,4], Ah Reum Jeong[1], Eun Soo Lee[5], Seung-Kuy Cha[6], Choon Hee Chung[5], Eun Young Lee[1,2,4] *

1 Department of Internal Medicine, Soonchunhyang University Cheonan Hospital, Cheonan, Korea,
2 Institute of Tissue Regeneration, College of Medicine, Soonchunhyang University, Cheonan, Korea,
3 Department of Pathology, Soonchunhyang University Cheonan Hospital, Cheonan, Korea, 4 BK21 Four Project, College of Medicine, Soonchunhyang University, Cheonan, Korea, 5 Department of Internal Medicine, Yonsei University Wonju College of Medicine, Wonju, Korea, 6 Department of Physiology, Yonsei University Wonju College of Medicine, Wonju, Korea

* eylee@sch.ac.kr

**Data Availability Statement:** All relevant data are within the paper and its Supporting Information files.

**Funding:** This work was supported by the National Research Foundation (NRF) of Korea grant funded

## Abstract

The anti-aging gene, klotho, has been identified as a multi-functional humoral factor and is implicated in multiple biological processes. However, the effects of klotho on podocyte injury in diabetic nephropathy are poorly understood. Thus, the current study aims to investigate the renoprotective effects of klotho against podocyte injury in diabetic nephropathy. We examined lipid accumulation and klotho expression in the kidneys of diabetic patients and animals. We stimulated cultured mouse podocytes with palmitate to induce lipotoxicity-mediated podocyte injury with or without recombinant klotho. Klotho level was decreased in podocytes of lipid-accumulated obese diabetic kidneys and palmitate-treated mouse podocytes. Palmitate-treated podocytes showed increased apoptosis, intracellular ROS, ER stress, inflammation, and fibrosis, and these were significantly attenuated by klotho administration. Klotho treatment restored palmitate-induced downregulation of the antioxidant molecules, Nrf2, Keap1, and SOD1. Klotho inhibited the phosphorylation of FOXO3a, promoted its nuclear translocation, and then upregulated MnSOD expression. In addition, klotho administration attenuated palmitate-induced cytoskeleton changes, decreased nephrin expression, and increased TRPC6 expression, eventually improving podocyte albumin permeability. These results suggest that klotho administration prevents palmitate-induced functional and morphological podocyte injuries, and this may indicate that klotho is a potential therapeutic agent for the treatment of podocyte injury in obese diabetic nephropathy.

## Introduction

Klotho was originally identified as an anti-aging gene [1], and it is predominantly produced in the distal convoluted tubules of the kidney and several other tissues, including the brain choroid plexus, pituitary gland, pancreases, and reproductive organs [2]. There are two distinct

by the Korea government (MSIT) (2020R1A2C2003438, 2019M3E5D1A02069071, 2018R1A2B6005360) and the Ministry of Education (2018R1A6A3A11040860). It was also supported by Soonchunhyang University Research Fund. The authors declare no conflicts of interest.

**Competing interests:** The authors have declared that no competing interests exist.

forms of klotho, including a membrane form and a secreted form. The membrane form serves as a coreceptor for fibroblast growth factor 23 (FGF-23). It modulates phosphate metabolism [3]. The secreted form by alternative klotho gene splicing or by the cleavage of large transmembrane klotho extracellular domain acts as a hormonal factor with anti-oxidation [4], anti-inflammatory [5], and anti-senescent [6] properties.

The kidney, under normal physiological conditions, is the major regulator that helps maintain klotho levels [7, 8]. However, klotho was reported to decline and to be associated with renal insufficiency following kidney injuries, such as streptozotocin (STZ)-induced hypertensive diabetic nephropathy (DN), chronic kidney disease (CKD), or ischemia/reperfusion acute kidney injury [9, 10]. In addition, our recent studies demonstrated that soluble klotho deficiency in plasma is associated with albuminuria in the patients with diabetes, and serum and urinary klotho levels correlate strongly with renal fibrosis and podocyte foot process effacement [11–13].

Diabetic nephropathy is considered the most common cause of CKD, comprising about 20 to 40 percent of all patients with type 2 diabetes mellitus [14]. Recent evidence suggest that soluble klotho plays a renoprotective role in the experimental models of diabetic nephropathy [5, 15, 16]. However, most studies have been focused on the protective effect of klotho against tubulointerstitial injury because of the main site of klotho production. The podocyte is a highly differentiated cell type in the kidney glomerulus, and its loss and dysfunction result in slit diaphragm disruption and proteinuria of both clinical and experimental DN [17–19]. Few studies have shown klotho expression in podocytes and its renoprotective effects in diabetic podocytes [20, 21]. However, little is known about whether and how klotho protects podocytes against glomerular damage.

In this study, we investigated the protective effects of klotho against palmitate-induced podocyte injury.

## Materials and methods

### Ethics statement

All animal experiments were performed in accordance with the recommendations in the Guidelines for Proper Conduct of Animal Experiments and Related Activities in Academic Research Institutions. The protocol was approved by the Committee on the Ethics of Animal Experiments of Yonsei University Wonju College of Medicine (Protocol Number: YWC-110225-1 and YWC-140304-1). Daily inspections were performed to minimize animal suffering and mice with signs of disease or discomfort were euthanized by CO2 and cervical dislocation. Surgical tissue isolations were performed as terminal procedures under anesthesia as described below.

The Institutional Reviewed Board (IRB) of Soonchunhyang University Cheonan Hospital (Cheonan, Korea) approved this study. The IRB waived the need to obtain informed consent from the participants. The methods in this study conformed to the relevant guidelines and regulations.

### Animal model

We purchased diabetic *db/db* mice (n = 9) and nondiabetic *db/m* mice (n = 9) at 6 to 7 weeks of age from Jackson Laboratory (Bar Harbor, MA, USA). We purchased *LETO* (n = 9) and *OLETF* (n = 8) rats from Otsuka Pharmaceutical Co., Ltd. (Tokushima, Japan). We anesthetized all experimental animals at the age of 16 to 46 weeks with Zoletil (Virbac Laboratories, Carros, France) and xylazine hydrochloride (Rompun TS, Bayer AG, Leverkusen, Germany) by intraperitoneal injection.

## Human specimens

We obtained normal control kidney tissue from two patients who had undergone uninephrectomy for renal cell carcinoma. These samples were free of neoplastic and glomerular lesions. The nondiabetic control specimen was obtained from a patient with minimal change disease (MCD). We collected a diabetic kidney biopsy from five patients, which showed the typical pathology of diabetic nephropathy (S1 Table in S1 File). All renal biopsy specimens were histopathologically re-examined by one pathologist (JH Lee) to confirm the diagnosis and pathological features of glomerular injury including glomerular basement membrane thickening, podocyte foot process effacement, and global sclerosis.

Klotho expression in renal biopsy specimens was investigated by immunohistochemistry. The paraffin-embedded kidney tissues were cut into 4-μm thickness, deparaffinized in xylene, hydrated using an ethanol-deionized water series, and stained with klotho (LS-B6625; LifeSpan BioSciences, Seattle, WA, USA). Primary antibody binding was detected using Bond Polymer Refine Detection Kit (Leica, Wetzlar, Germany).

## *In vitro* cell culture

Dr. Peter Mundel kindly provided conditionally immortalized mouse podocytes [22]. We cultured the podocytes at 33˚C under permissive conditions in DMEM containing 10% FBS and 10 U/mL of mouse recombinant interferon-γ (Sigma-Aldrich, St. Louis, MO, USA) to enhance the expression of a thermosensitive T antigen. For differentiation, we cultured podocytes under nonpermissive conditions at 37˚C without interferon-γ for 14 days. We maintained these cells under serum-deprived conditions for 24 hours, treated them with 400 μM of palmitate (Pal) with or without 400 pM of recombinant human klotho protein (R&D systems, USA), oleate (Sigma-Aldrich, St. Louis, MO, USA), or NAC (Sigma-Aldrich) for 24 hours. We then harvested them for the next assay.

Dr. Moin A. Saleem (University of Bristol, Bristol, UK) generously provided human conditionally immortalized podocytes (AB8/23) [23]. We cultivated the human podocytes at 33˚C (permissive conditions) in an RPMI-1640 medium supplemented with 10% FBS and Insulin-Transferrin-Selenium-Pyruvate Supplement (ITSP; WelGENE Inc., Daegu, South Korea) to induce expression of a thermosensitive T antigen. For differentiation, we maintained podocytes at 37˚C (non-permissive conditions) without ITSP for 14 days. We grew immortalized human tubule cells in a DMEM/F12 medium containing 10% FBS.

## Klotho ELISA

α-klotho in culture media was analyzed using a mouse klotho ELISA kit (Cusabio, Houston, TX, USA) according to the manufacturer's protocol. Samples were analyzed in duplicate and were within the range of the standard curve (3.12–200 pg/mL).

## Oil Red O staining

We sliced an OCT-embedded frozen kidney tissue sample into 4-μm thick sections. We fixed the kidney samples and cells in 4% formalin for 15 minutes, washed them with PBS, and stained them with Oil Red O (Sigma-Aldrich) working solution for 30 minutes. After removing the Oil Red O solution, we immediately washed the samples with 60% isopropanol for 5 seconds and then counterstained them for 5 minutes. We observed the histological changes with a microscope (Leica, Wetzlar, Germany).

## Immunofluorescence

We fixed podocytes that had been cultured on collagen-coated coverslips for 14 days with 4% paraformaldehyde, permeabilized with 0.25% Triton X-100, blocked with 1% BSA, and immunolabeled with FITC-phalloidin (Sigma-Aldrich, St. Louis, MO, USA), ZO-1 (Invitrogen, Carlsbad, CA, USA) and paxillin (Invitrogen), as well as FOXO3a (Cell Signaling Technology, Danvers, MA, USA). Subsequently, we incubated the cells with Alexa 594-conjugated anti-rabbit antibody (Invitrogen) for 2 hours at RT. The images were collected using an LSM 510 META laser-scanning confocal microscope (Carl Zeiss Microimaging, Thornwood, NY, USA) at the Soonchunhyang Biomedical Research Core Facility of Korea Basic Science Institute (KBSI).

## Quantitative real-time PCR

Total RNA was isolated from the kidney cortex and cultured mouse podocytes using TRIzol (Sigma). cDNA was synthesized from 0.5–1 µg of RNA with a ReverTraAce® qPCR RT master Mix (TOYOBO, Japan) according to the manufacturer's protocol. To evaluate mRNA expression, cDNA was amplified using SYBR Green PCR Master Mix (TOYOBO). The primer pairs for mklotho, mMCP-1, mBax, mBcl2, mChop, and rklotho were as follows. mklotho: 5'-agc aca ggt ttg cgt agt ct-3' (forward) and 5'-caa tgg ctt ccc tcc ttt ac-3' (reverse); mMCP-1: 5'-ctg gat cgg aac caa atg ag-3' (forward) and 5'-cgg gtc aac ttc aca ttc aa-3' (reverse); mBax: 5'- gga tgc gtc cac caa gaa g-3' (forward) and 5'- caa agt aga ggg caa cca c-3' (reverse); mBcl2: 5'- tgt ggt cca tct gac cct cc-3' (forward) and 5'- aca tct ccc tgt tga cgc tct-3' (reverse); mChop: 5'- tgt ggt cca tct gac cct cc-3' (forward) and 5'- aca tct ccc tgt tga cgc tct-3' (reverse); mRPL13A: 5'-cga tag tgc atc ttg gcc ttt-3' (forward) and 5'- cct gct ctc aag gtt gtt-3' (reverse); rklotho: 5'-cgt gaa tga ggc tct gaa ag-3' (forward) and 5'-gag cgg tca cta agc gaa ta-3' (reverse). Real-time PCR reactions were carried out on CFX connect™ (Bio-Rad) and data analysis was performed following $\Delta\Delta C_T$ method. Data were normalized by β-actin or rPL13A mRNA levels in the same sample.

## TUNEL assay

We detected DNA fragmentation, which is one of the later steps in apoptosis, by using the Apo-BrdU in situ DNA fragmentation kit (BioVision, Milpitas, CA, USA). We grew the podocytes on glass cover slips, fixed them in 4% paraformaldehyde in PBS (pH 7.4), and permeabilized them with 0.1% Triton X-100 in 0.1% sodium citrate. The commercial assay was performed in accordance with the manufacturer's specifications. We labeled the apoptotic cells with exposed 3'-hydroxyl DNA ends with brominated deoxyuridine triphosphate nucleotides (Br-dUTP). We then used the FITC labeled anti-BrdU mAb to stain the apoptotic cells. We visualized the cells by using the LV10i inverted confocal microscope (Olympus, Tokyo, Japan).

## Measurement of Reactive Oxygen Species (ROS) generation

We detected intracellular ROS generation by using a 2'-7' dichlorofluorescein diacetate (CM-H2DCFDA; Molecular Probes, Eugene, OR, USA) fluorescent probe. We loaded the podocytes onto a glass dish with 5 µM CM-H2DCF-DA for 20 minutes at 37˚C. We washed the excess dye out with PBS. We measured the fluorescence intensity by using an LV10i inverted confocal microscope.

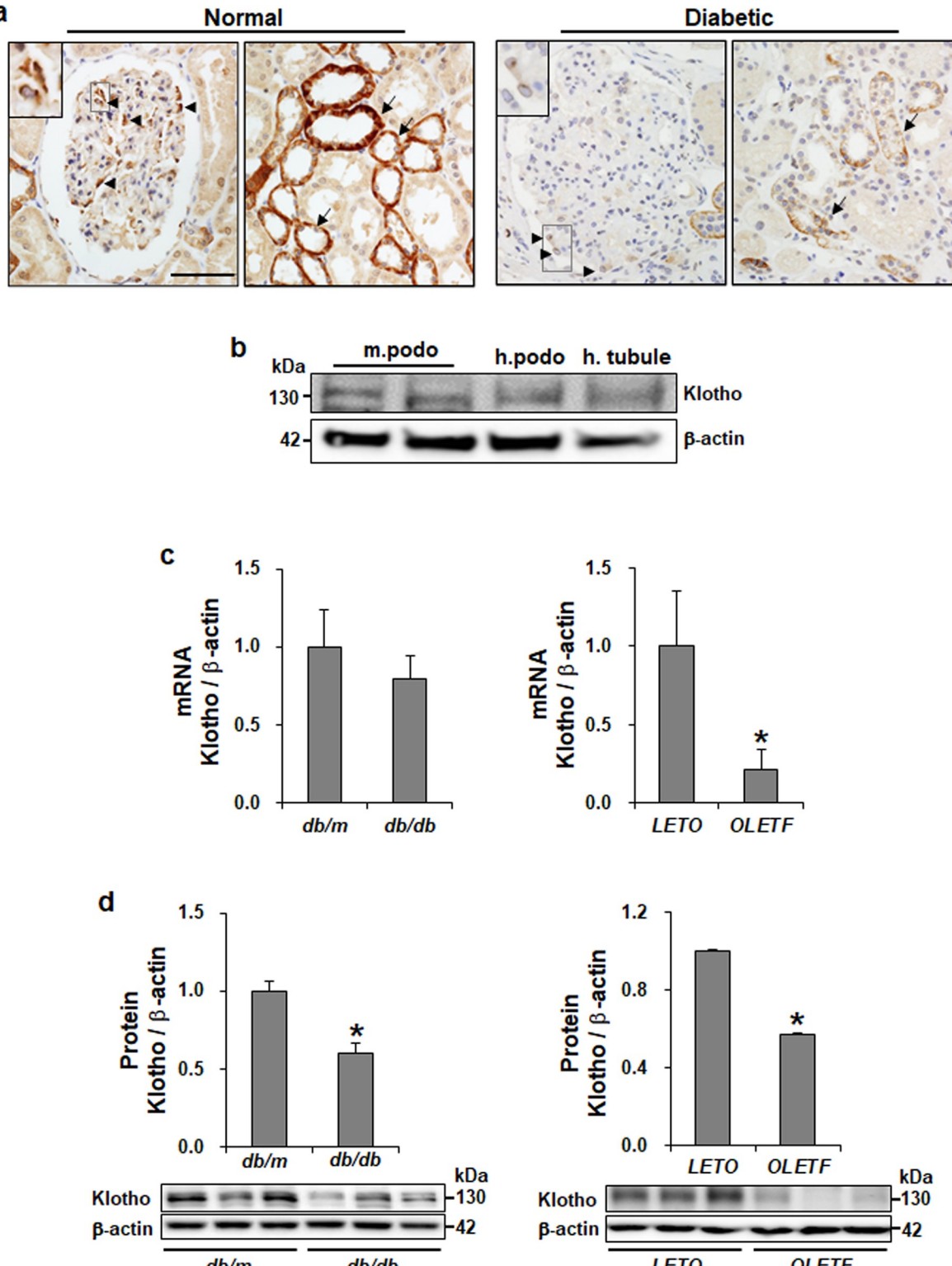

**Fig 1. Diabetic kidney shows decreased klotho expression.** (a) Representative images of the kidney from normal and diabetic biopsies showing the decreased klotho expression in diabetic podocytes by IHC. Arrowheads and arrows indicated the podocytes and distal tubules expressing klotho. Boxes are enlarged images of representative klotho expression in podocytes and tubules. magnification 40x; scale bar = 50 μm. (b) Verification of klotho expression in immortalized mouse and human podocytes compared to human tubule cells by Western blot. (c) Klotho gene was quantified by real-time qPCR analysis of total mRNA isolated from diabetic and nondiabetic kidneys. (d)

Western blot demonstrating decreased klotho in diabetic *db/db* mice (n = 9) and *OLETF* rats (n = 8). Data is presented as the mean ± SEM.
* P < 0.05 versus nondiabetic *db/m* mice and *LETO* rats.

## Western blotting

We homogenized the cultured podocytes and kidney cortex in PRO-PREPTM protein extraction solution (iNtRON Biotechnology, Seoul, Korea) containing a protease inhibitor cocktail (Roche Diagnostics GmbH, Mannheim, Germany). We determined the protein concentration by using a Bradford assay (Bio-Rad, Hercules, CA, USA). We loaded equal amounts of protein samples per lane, separated them by SDS-PAGE, and transferred them onto a PVDF membrane (Millipore, Billerica, MA, USA). We blocked the membrane by using 5% non-fat dry milk, followed by primary antibody incubation at 4°C overnight: klotho (R&D system, Minneapolis, USA), nephrin (Progen, Heidelberg, Germany), Nrf2 (Santa Cruz Biotechnology, Santa Cruz, CA, USA), Bcl2 (Santa Cruz Biotechnology), Bax (Santa Cruz Biotechnology), cleaved caspase 3 (Cell Signaling Technology), Bip (Santa Cruz Biotechnology), TNFR2 (Santa Cruz Biotechnology), TLR4 (Santa Cruz Biotechnology), TGF-β1(Santa Cruz Biotechnology), ATF4 (Santa Cruz Biotechnology), keap1 (Cusabio), SOD1 (Cusabio), IL-6 (Cusabio), TNF-α (Cell Signaling Technology), TRPC6 (Abcam, Cambridge, MA, USA), Fibronectin (Dako A/S, Glostrup, Denmark), p-FOXO3a (Cell Signaling Technology), FOXO3a (Cell Signaling Technology), MnSOD (Abcam), and VEGF (Invitrogen). Following primary antibody binding, we incubated the membranes with horseradish peroxidase (HRP)-conjugated secondary antibody. We visualized the bands with a ChemiDoc TM XRS+ (Bio-Rad, Hercules, CA, USA) imaging system using a Luminata Forte enhanced chemiluminescence solution (Millipore).

## Statistical analysis

We present the experimental values as a mean ± SEM or SD. The statistical analysis was performed using a two-tailed unpaired Student's *t*-test, and one-way ANOVA. *P value less than 0.05 was considered significant.*

## Results

### Decreased klotho expression in the glomeruli of obese diabetic nephropathy

To determine whether podocytes in the glomerulus are one of the sources to express klotho, we analyzed the distribution of klotho in human kidney using immunohistochemistry. Consistent with the previous result, we observed the klotho in the distal tubules being surrounded by normal tubulointerstitium. Podocytes in the normal glomerulus showed klotho expression as a cytoplasmic pattern, whereas the expression level of klotho was decreased in diabetic podocytes and tubules (Fig 1A). As shown in Fig 1B, we verified the expression of klotho in both immortalized mouse and human podocytes using Western blots. Proteinuria is a common consequence of kidney disease and is considered a marker of the severity of the disease processes. In *db/db* and *OLETF* diabetic mice with increased albuminuria, renal expression of klotho was significantly decreased at RNA and protein levels compared to the control (Fig 1C and 1D).

### Decreased klotho expression in palmitate-treated mouse podocytes

Renal lipid accumulation leads to glomerular damage and produces kidney dysfunction. To determine the renal lipid accumulation in obese diabetic kidney and mouse podocytes, we

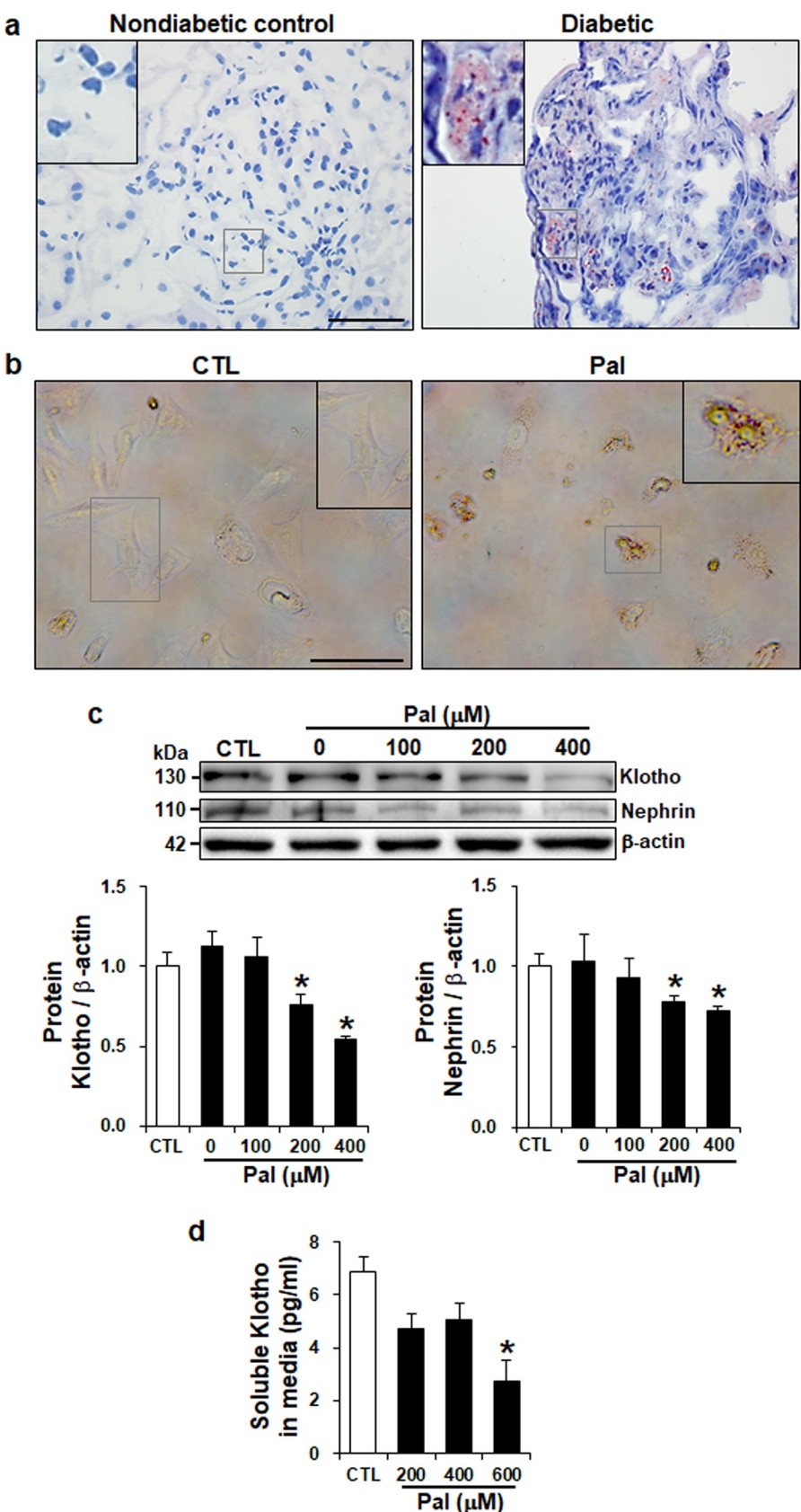

**Fig 2. Increased lipid accumulation and decreased klotho expressions in palmitate-treated mouse podocytes.** (a and b) Representative Oil-Red O staining from diabetic human kidney biopsies (a) and palmitate-treated mouse podocytes (b) demonstrating the increased lipid accumulation. Boxes are enlarged images of representative lipid droplets. magnification 20x; scale bar = 50 μm and 100 μm. (c) Western blot showing the decreased expression of klotho and nephrin proteins by palmitate treatment. Podocytes that were not treated with palmitate or BSA were marked as 0 μM. (d) ELISA assay demonstrating decreased soluble klotho by palmitate treatment. Culture media was collected from podocytes treated by palmitate in a dose-dependent manner. Data is presented as the mean ± SD. Similar results were obtained in two independent experiments. $^*P < 0.05$ versus BSA-treated control (CTL).

visualized the lipid contents by using Oil-Red O staining. The kidney from diabetic patients showed increased lipid accumulation in glomeruli compared to nondiabetic control (Fig 2A), and podocytes treated with palmitate accumulated intracellular lipid droplet formation (Fig 2B). To evaluate the expression levels of klotho and slit diaphragm protein, nephrin, we incubated podocytes with palmitate in a dose-dependent manner and harvested them at 24 hours following treatment. Western blot analysis showed that klotho and nephrin protein levels were significantly decreased in palmitate-treated podocytes compared to BSA control (Fig 2C). Furthermore, stimulation of high glucose (HG) and advanced glycation end-products (AGE), and co-treatment of HG with palmitate reduced klotho expressions at RNA and protein levels, respectively (S1A and S1B Fig in S1 File). To investigate whether palmitate decreased soluble klotho in mouse podocytes, an ELISA assay was performed using supernatant collected from mouse podocytes. Consistent with our previous study demonstrating that soluble klotho was negatively correlated with albuminuria in type 2 diabetic nephropathy [11, 12], soluble klotho in culture media was decreased by palmitate treatment compared to control (Fig 1D).

## Recombinant klotho inhibits palmitate-induced apoptosis in mouse

To examine whether klotho could prevent palmitate-induced cell death, we stimulated podocytes with palmitate, with or without recombinant klotho (rKL). As shown in Fig 3A, decreased cell viability by palmitate was significantly ameliorated by the rKL treatment. Apo-BrdU TUNEL assay showed the increased incorporation with DNA strand breaks in palmitate-treated podocytes than in the control, whereas rKL treatment significantly decreased BrdU-FITC expression, as observed in oleate-treated cells (Fig 3B). Further, cleaved caspase-3 and Bax, which are known as apoptosis-related molecules, were significantly increased in palmitate-treated podocytes as compared to the control cells, whereas the increased expression levels of these proteins were significantly attenuated by rKL. Furthermore, Bcl2 decreased by palmitate treatment was restored by KL (Fig 3C and 3D). This data indicates that klotho prevents palmitate-induced cytotoxicity in podocytes.

## Klotho inhibits palmitate-induced oxidative stress and ER stress

Next, we examined the effect of klotho on ROS production and ER stress. As shown in Fig 4A and 4B, intracellular ROS was increased in palmitate-treated podocytes compared to control, whereas the increased ROS production was significantly ameliorated by rKL treatment as shown in ROS scavenger NAC-treated podocytes and oleate. In addition, the ER stress markers, such as Bip, ATF4, and Chop increased in palmitate-treated podocytes, were significantly decreased by rKL as shown in NAC-treated cells (Fig 4C and 4D).

## Klotho induces antioxidant activity through FOXO3a in palmitate-treated mouse podocytes

To examine the antioxidant activity of klotho on palmitate-induced oxidative stress, we investigated Nrf2, which plays a critical role in the response against oxidative stress. After the

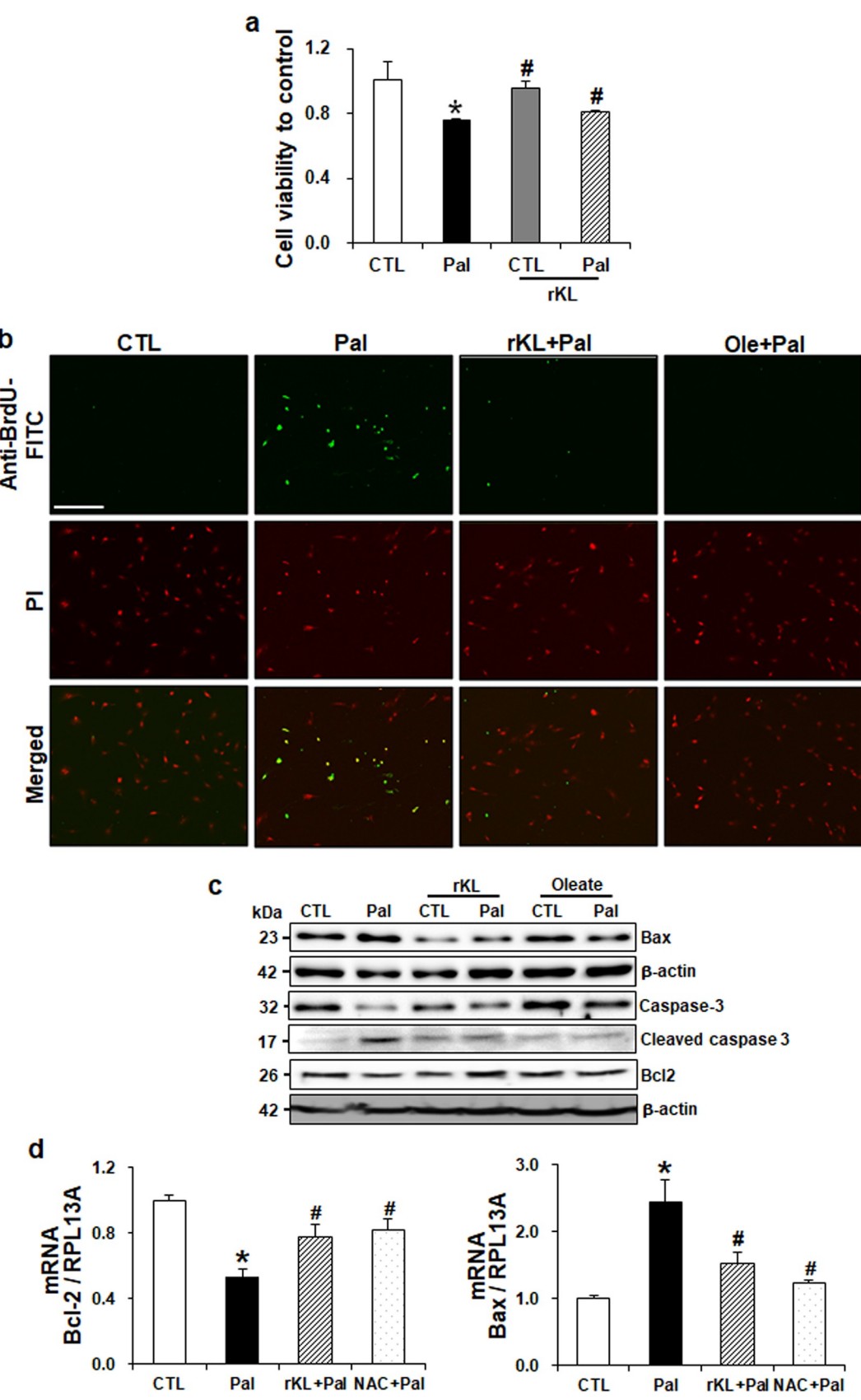

**Fig 3. Klotho protects podocytes from palmitate-induced cell death.** Mouse podocytes were stimulated by palmitate with or without treatment of rKL for 24 hours. (a) Cell cytotoxicity was assessed by MTT assay ($n = 6$). (b) Podocytes grown on coverslips were fixed with 4% PFA and performed Apo-BrdU in situ DNA fragmentation assay. Cells were stained with propidium iodide (PI, red) and the Br-dUTP/TdT enzyme (green). Magnification 10x; bar = 200 μm. (c) Western blot and (d) real-time qPCR (n = 3) demonstrating the restored expression of apoptosis-related molecules by rKL. Consistent with rKL, oleate and ROS scavenger NAC restored the palmitate-induced apoptosis. Data are presented as the mean ± SD. The experiments were repeated twice. *P < 0.05 compared to control (CTL). #P < 0.05 compared to palmitate (Pal).

treatment with palmitate, the expression levels of Nrf2 and keap1 were downregulated and Nrf2 down-stream target SOD1 was also decreased. However, rKL treatment restored the decreased expression of these proteins (Fig 5A). Next, to investigate the underlying molecular mechanism for the protective role of klotho, we checked the expression of antioxidant enzyme induced by FOXO3a activity. As shown in Fig 5B, western blots with whole-cell lysates revealed that palmitate treatment increased FOXO3a phosphorylation, whereas rKL inhibited FOXO3a phosphorylation. Also, rKL restored MnSOD expression decreased by palmitate. Fig 5C showed the klotho-induced FOXO3a nuclear translocation by immunostaining assay. Palmitate-treated podocytes showed decreased levels of FOXO3a in the nucleus compared to control cells (Fig 5C). However, treatment of rKL restored FOXO3a levels in the nucleus compared to palmitate-treated cells indicating that palmitate-mediated FOXO3a phosphorylation inhibits translocation of FOXO3a into the nucleus. This data suggests that klotho protects podocyte injury against palmitate-induced oxidative stress by enhancing the activity of antioxidants.

## Klotho suppresses the expression of inflammatory cytokines and fibrotic markers

Along with oxidative stress, inflammation and fibrosis have been associated with the progression of DN. The expression levels of inflammatory cytokines, tumor necrosis factor (TNF)-α, TNFR2, TLR4, and IL-6 at protein level, and MCP-1 at RNA level were increased in palmitate-treated podocytes, but rKL treatment attenuated these increased expression levels (Fig 6A and 6B). As shown in Fig 6C, the expression levels of fibronectin and TGF-β1, the profibrotic factors, were increased in the palmitate-treated podocytes. In addition, palmitate treatment exhibited an increased expression of VEGF associated with pathogenesis of diabetic retinopathy. However, rKL treatment exhibited a decreased expression of these proteins.

## Klotho attenuates palmitate-induced actin cytoskeleton disruption and albumin permeability

To assess whether klotho could restore palmitate-induced cytoskeleton rearrangement and changes in cell-cell junctions, we subjected podocytes to immunofluorescence. As shown in Fig 7A, palmitate disrupted the uniformly organized actin stress fibers throughout the podocyte cytoplasm, whereas rKL treatment restored these changes. The expression of ZO-1 at the cell junction was reduced in palmitate-treated podocytes as compared to the control cells. Treatment with rKL restored the ZO-1 levels, as did the F-actin. Furthermore, rKL treatment restored the decreased expression of nephrin and the increased TRPC6 expression (Fig 7B). To evaluate the effect of klotho on podocyte filtration barrier function, we tested the albumin permeability. We observed a significant increase in albumin permeability in palmitate-treated podocytes, whereas rKL treatment attenuated albumin leakage (Fig 7C). This data suggests that klotho could improve the podocyte actin cytoskeleton and function.

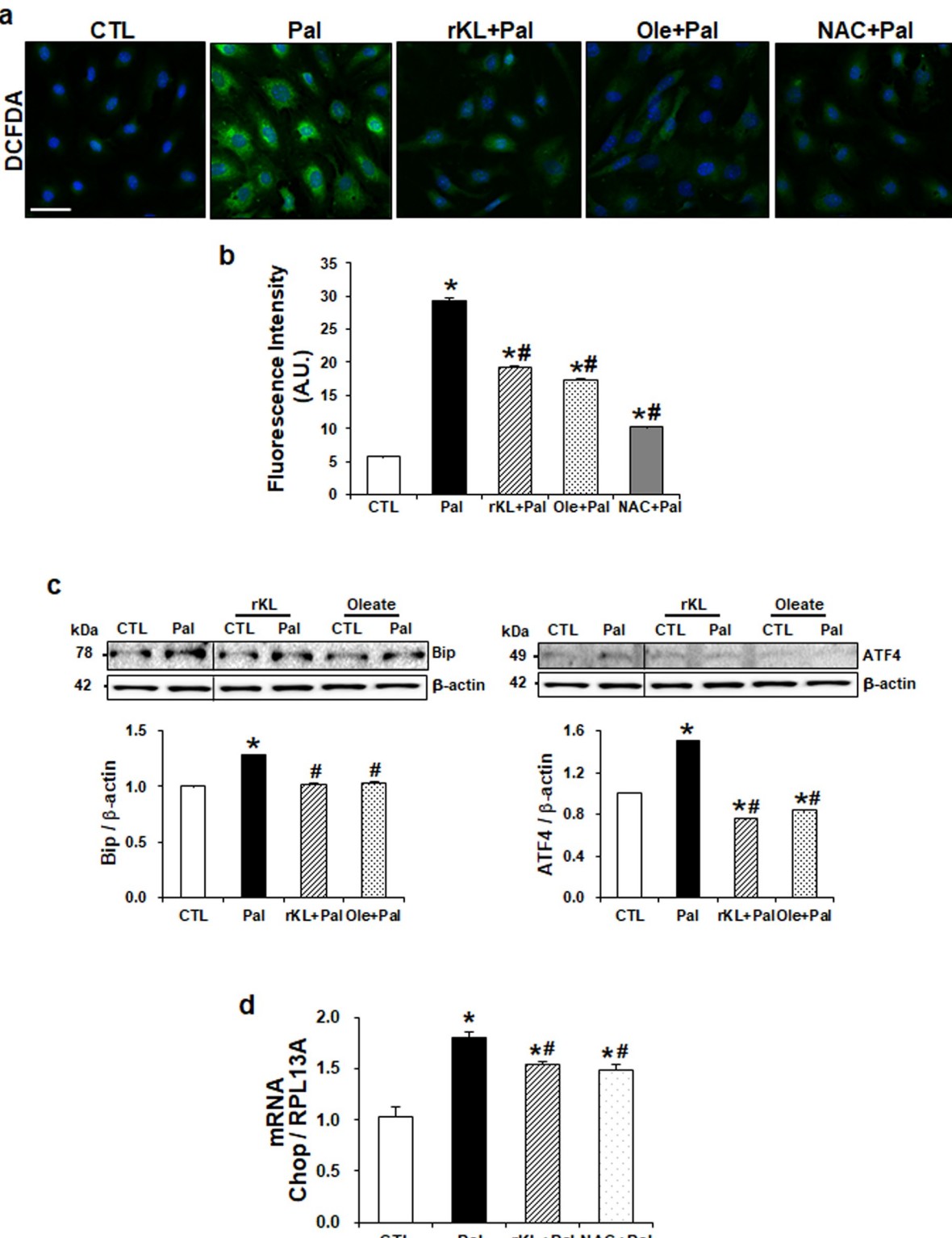

**Fig 4. Klotho protects mouse podocytes from palmitate-induced oxidative stress and ER stress.** (a) Representative images of ROS measurements in podocytes loaded with 2'7'-dichlorofluorescein (DCF) dye. Magnification 40x; bar = 40 μm. (b) The intensity of DCF was quantified by Image J software. (c) Western blot and (d) real-time qPCR (n = 3) showing decreased expression of ER stress markers by rKL. The blots were cropped from different parts of the same gel. Data are expressed as mean ± SEM for three experiments.

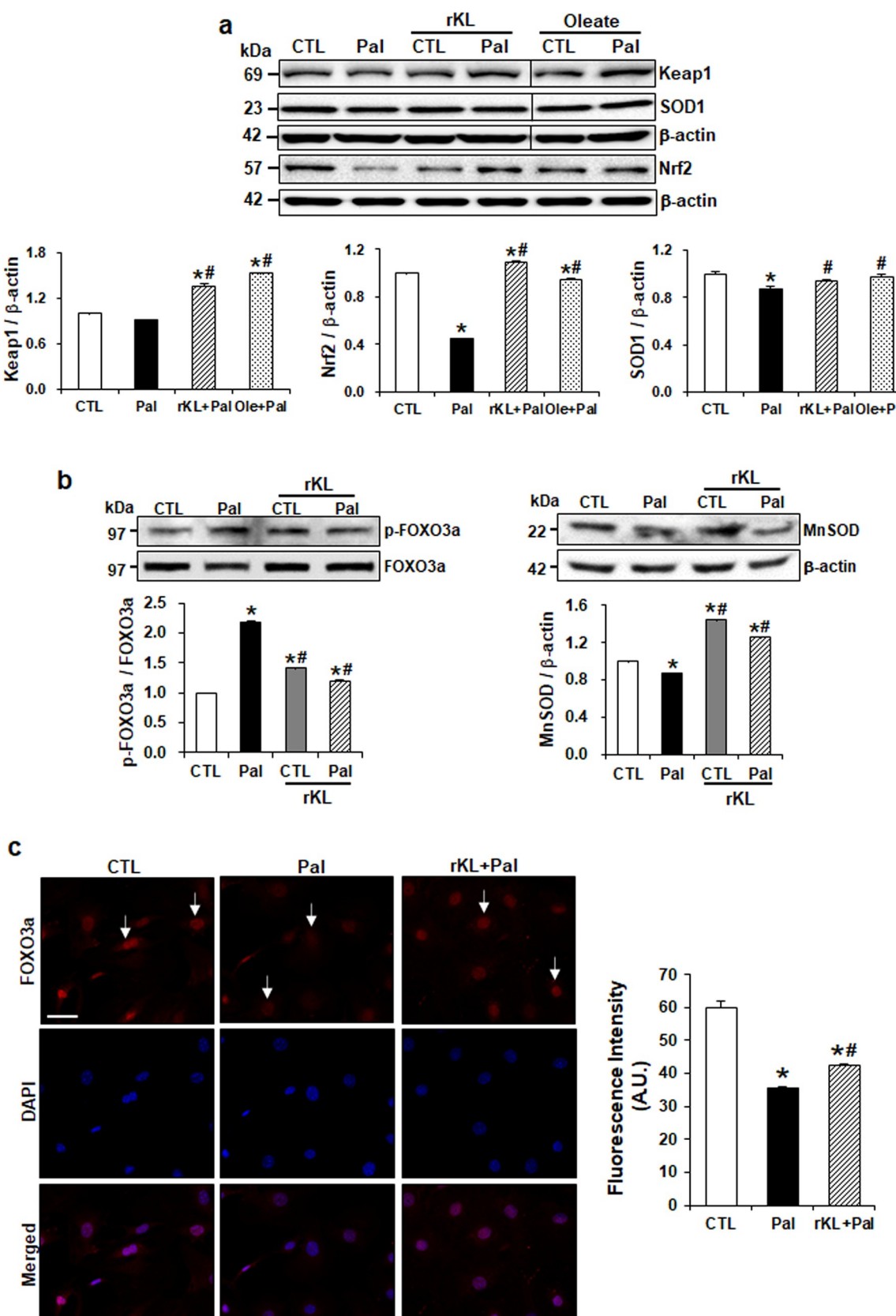

**Fig 5. Klotho induces antioxidant activity by regulating the FOXO3a in palmitate-treated mouse podocytes.** (a) Western blots demonstrating the restored expression of anti-oxidative proteins by rKL. The blots were cropped from different parts of the same gel. (b) Western blots measuring the protein expression of phosphorylated FOXO3a (p-FOXO3a) and MnSOD. All proteins were normalized to β-actin or total (t) protein controls. (c) Immunofluorescence staining showing klotho-induced FOXO3a nuclear translocation. Arrows indicated FOXO3a expression in the nucleus. The intensity of DCF was quantified by Image J software. Magnification 40x; scale bar = 40 μm. Data are expressed as mean ± SEM for three experiments. *$P < 0.05$ compared to control (CTL). #$P < 0.05$ compared to palmitate (Pal).

## Discussion

We have demonstrated that klotho in glomerular podocytes was decreased in the lipid-accumulated obese diabetic kidney and palmitate-treated podocytes. The administration of rKL protein could protect podocytes from injury under obese diabetic mimic conditions by inhibiting intracellular ROS, ER stress, and apoptosis. rKL inhibited the phosphorylation of FOXO3a induced by palmitate, and it promoted its nuclear translocation. Therefore, rKL increased MnSOD and antioxidants expression during palmitate-induced toxicity in podocytes. Furthermore, rKL restores palmitate-mediated inflammation, fibrosis, disruption of the actin cytoskeleton, and increased albumin permeability. These findings suggest that rKL restores palmitate-mediated functional and morphological podocyte injuries, which indicates that klotho has a protective effect on the glomerular injury of obese DN.

Abnormal lipid metabolism and accumulated lipids in non-adipose tissue causes various types of cellular and organ damage in several chronic diseases, including DN. Free fatty acid (FFA) is the major pathogenic mediator in the development of diabetes mellitus and its complications [24]. Palmitate is the most abundant circulating saturated FFA in human and rodent plasma [25, 26] and impairs insulin signaling and increases ER stress, whereas monounsaturated fatty acid oleate has a protective effect by restoring palmitate-induced defects at insulin signaling [27]. Type 2 diabetic patients showed 1.5- and 3-fold higher plasma palmitate levels compared with healthy subjects [28]. Accumulated renal lipids lead to glomerular damage and produce dysfunction in podocytes that maintain a glomerular filtration barrier. Consistent with previous studies showing lipid accumulation in podocytes and human DN [24, 29], in the present study, lipid deposition was observed in obese diabetic human kidneys and in palmitate-stimulated podocytes. In many studies, a renal klotho expression is decreased in experimental animal models and in the patients with diabetic nephropathy [9, 10], and plasma soluble klotho is negatively correlated with the progression of nephropathy with type 2 diabetic patients [11–13]. Further, hyperglycemia and angiotensin II inhibited klotho gene expression [30, 31]. Consistently, we found that glomerular podocyte is another source of klotho expression along with renal tubules, and klotho expression is decreased from lipid-accumulated obese diabetic kidney and palmitate-induced podocyte injury.

Production and function of klotho are reduced in obese diabetes, and rKL administration can make up for those losses. Increasing evidence has shown that klotho administration or overexpression can be renoprotective against glomerular injury. Deng *et al.* showed that the klotho transgene ameliorates kidney hypertrophy and glomerular injury in STZ-induced DM [15]. In addition, Kim *et al.* observed that klotho attenuates proteinuria by suppressing the TRPC6 channel in podocytes [20]. Furthermore, Oh *et al.* investigated whether klotho exerts a renoprotective effect against glomerular injury in diabetes, and observed that administration of klotho has a protective effect on glomerular hypertrophy via a cell cycle-dependent manner and decrease albuminuria in DM [21]. However, the renoprotective effect of klotho in lipotoxicity-induced obese diabetic kidney disease remains unclear.

The accumulation of palmitate induces oxidative stress affecting protein misfolding [32]. Excessive production of ROS induced by palmitate participates in ER $Ca^{2+}$ depletion and ER

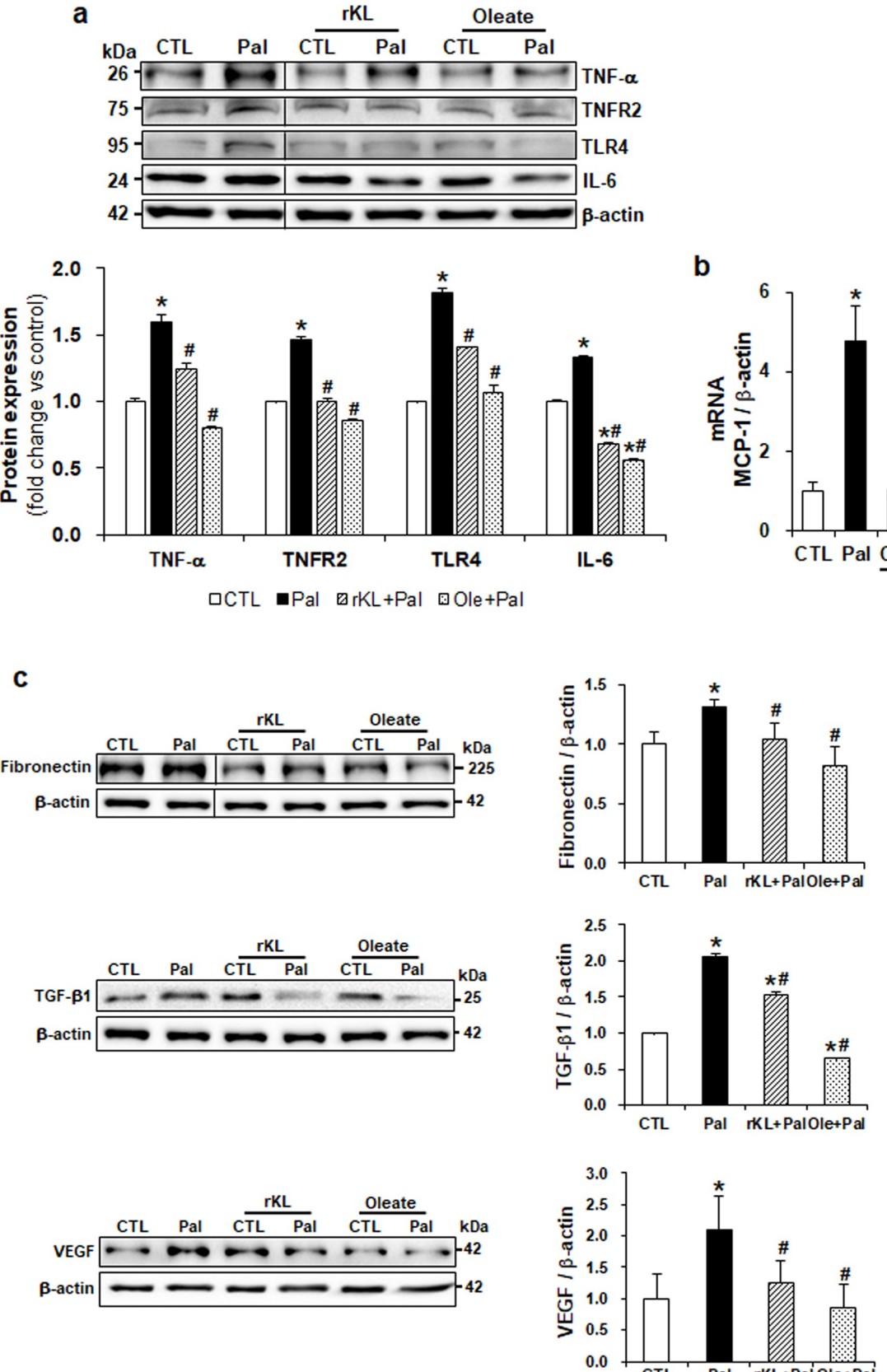

**Fig 6. Klotho ameliorates palmitate-induced inflammatory chemokines and fibrosis in mouse podocytes.** (a) Western blots demonstrating the restored expression of inflammatory chemokines. (b) Real-time qPCR demonstrating decreased MCP-1 at the RNA level by rKL treatment. (c) Western blots showing restored expression of fibrosis-related molecules by treatment of rKL. The blots were cropped from different parts of the same gel. Data are presented as the mean ± SD. The experiments were repeated three times. *P < 0.05 compared to control (CTL). #P < 0.05 compared to palmitate (Pal).

stress-mediated cell death [33]. Our study showed that the anti-ROS function of klotho restores the upregulated expression of ER stress-associated proteins and cell death in palmitate-mediated podocyte injury. Increased oxidative stress occurs by an imbalance between ROS production and the antioxidant defense system in DM [34]. The overproduction of ROS in mitochondria under pathogenic conditions results in detrimental consequences, leading to impaired cell function and death [35, 36]. Nrf2 is a transcriptional factor that regulates the cellular antioxidant response by binding to the antioxidant response element (ARE) and appears to counteract renal damage in diabetes [37, 38]. Klotho induces Nrf2-mediated antioxidant defenses in several models [39, 40]. In our study, we observed that klotho increased Nrf2 and SOD1 proteins in palmitate-treated podocytes.

The FOXO transcription factor is an important regulator of longevity and cancer by regulating target genes associated with cellular differentiation, oxidative stress resistance and nutrient shortage [41]. Phosphorylation modification regulates FOXO3a activity through a cytoplasmic-nuclear shuttle mechanism. The PI3K-AKT signaling pathway regulates FOXO3a activity through phosphorylation at three conserved residues (Thr 32, Ser 253, and Ser 315). This phosphorylation excludes FOXO3a from the nucleus and induces binding to the 14-3-3 nuclear export proteins, promoting its cytoplasmic accumulation and inhibiting the transactivating activity of FOXO3a [42]. Yamamoto *et al.* demonstrated that klotho reduces FOXO phosphorylation and promotes its nuclear translocation [4]. Lim *et al.* showed klotho's protective role during tacrolimus-induced oxidative stress via FOXO3a-mediated MnSOD expression [43]. In the present study, we found that klotho decreases palmitate-induced FOXO3a phosphorylation resulting in its nuclear translocation and enhances antioxidant expression, including MnSOD. Taken together, this data suggests that klotho protects podocyte dysfunction against palmitate, which might be dependent on the FOXO3a-mediating antioxidant defense system.

Growing evidence suggests that elevated pro-inflammatory cytokines, along with oxidative stress, have been involved in the pathogenesis of diabetes mellitus and DN [44, 45]. The mechanism of lipotoxicity directly or indirectly involved the activation of inflammatory and profibrotic responses [46, 47]. The enhanced expression of IL-6, TNF-α, and IL-1 in DN is related to proteinuria and the progression of DN. These inflammatory cytokines can activate the production of VEGF [48]. Klotho has been reported to exhibit anti-inflammatory activity under several pathological conditions [49, 50]. In our experiment, we showed that klotho administration prevented increased inflammatory cytokines, TNF-α, IL-6, and TNFR2 and MCP-1, and also increased fibrosis-related molecules, fibronectin, and TGF-β1 in the palmitate-stimulated podocytes. These results suggested that klotho administration has a protective effect on inflammation and fibrosis mediated by lipotoxicity.

Proteinuria is a characteristic milestone in DN and is considered a marker of the disease process's severity. The structure of the actin cytoskeleton in podocytes is important in maintaining a proper glomerular filtration barrier function [51, 52]. Palmitate-induced cytoskeleton rearrangements in podocytes are associated with ER $Ca^{2+}$ release mediated by phospholipase C (PLC) activation [20]. In our previous study, we observed that TRPC6, a downstream target of Ang II receptor signaling, is associated with the rearrangement of the actin cytoskeleton and albumin permeability via $Ca^{2+}$ influx [53]. Some studies have shown that soluble klotho

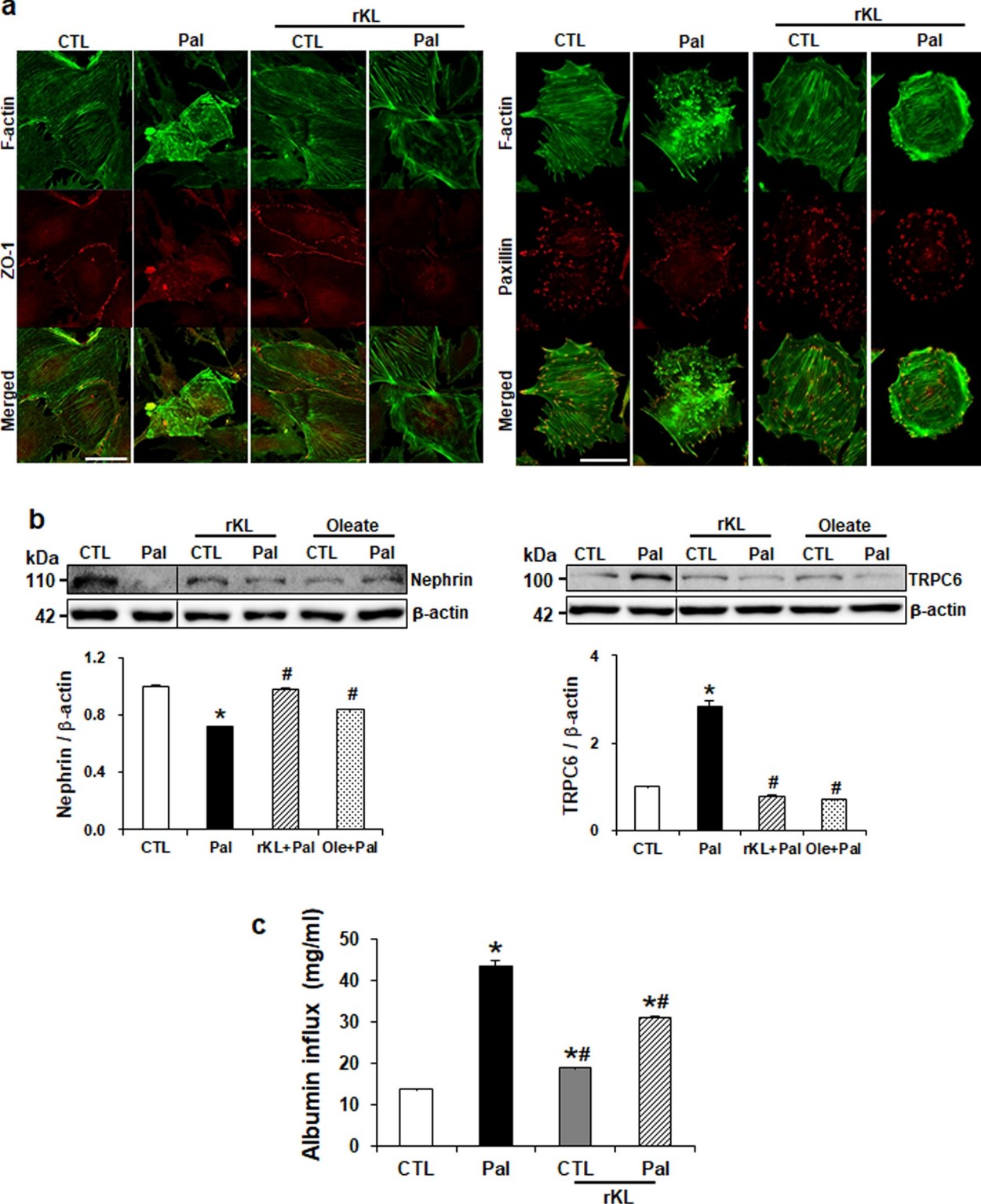

**Fig 7. Klotho restores palmitate-induced actin cytoskeleton disruption and impaired filtration barrier function of mouse podocytes.** (a) Representative morphologic changes of F-actin in palmitate-treated podocytes with or without rKL. Magnification 40x; scale bar = 50 μm. (b) Western blots demonstrating the restored expression of nephrin (left) and TRPC6 (right) by rKL. The blots were cropped from different parts of the same gel. (c) Podocyte permeability to albumin was measured. Podocytes on transwell filter chambers were treated with palmitate with or without rKL. Bovine serum albumin-containing medium (40 mg/ml) was added into the lower chambers, and then the upper chambers were sampled. Data

is presented as the mean ± SD. The experiments were repeated three times. *P < 0.05 compared to control (CTL). #P < 0.05 compared to palmitate (Pal).

prevents renal damage [54] and protects against actin cytoskeletal reorganization and albumin leakage in podocytes by suppressing TRPC6 overexpression [20]. In this study, we found that the administration of klotho prevents the palmitate-induced arrangement of the actin cytoskeleton, decreased expression of nephrin, and albumin permeability. Also, klotho prevents palmitate-induced TRPC6 overexpression.

In conclusion, we demonstrated that klotho is expressed in glomerular podocytes and that lipid-accumulated podocytes decrease klotho expression. Klotho administration protects against oxidative stress, inflammatory responses, fibrosis, cytoskeleton changes, and cell death in palmitate-induced podocyte injury, eventually improving the podocyte filter function. These renoprotective effects are strongly associated with its ability to inhibit palmitate-induced oxidative stress by inhibiting phosphorylation of FOXO3a mediating antioxidant defense system. Our observations strongly suggest that klotho may be a potential therapeutic agent to treat diabetic podocytes.

## Supporting information

**S1 Raw images.**
(PDF)

**S1 File.**
(DOCX)

## Author Contributions

**Conceptualization:** Jeong Suk Kang, Eun Young Lee.

**Data curation:** Jeong Suk Kang.

**Formal analysis:** Jeong Suk Kang.

**Funding acquisition:** Jeong Suk Kang, Choon Hee Chung, Eun Young Lee.

**Investigation:** Jeong Suk Kang, Eun Young Lee.

**Methodology:** Jeong Suk Kang, Seung Seob Son, Ji-Hye Lee, Seong Woo Lee, Ah Reum Jeong.

**Project administration:** Jeong Suk Kang, Eun Young Lee.

**Resources:** Eun Soo Lee.

**Supervision:** Seung-Kuy Cha, Eun Young Lee.

**Validation:** Ji-Hye Lee, Eun Young Lee.

**Visualization:** Jeong Suk Kang, Eun Young Lee.

**Writing – original draft:** Jeong Suk Kang.

**Writing – review & editing:** Seung-Kuy Cha, Choon Hee Chung, Eun Young Lee.

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
