## [Decision Letter · Decision Letter 0]

6 Nov 2020

PONE-D-20-26221

Klotho has protective effects on podocyte injury under diabatic conditions

PLOS ONE

Dear Dr. Lee,

Thank you for submitting your manuscript to PLOS ONE. After careful consideration, we feel that it has merit but does not fully meet PLOS ONE’s publication criteria as it currently stands. Therefore, we invite you to submit a revised version of the manuscript that addresses the points raised during the review process.

Your manuscript was reviewed by two experts and both of them provided many major comments. Please address those comments as appropriate. 

We look forward to receiving your revised manuscript.

Kind regards,

Partha Mukhopadhyay, Ph.D.

Academic Editor

PLOS ONE

Journal Requirements:

2.Thank you for including your ethics statement:

Animal Research

We carried out all experiments in conformity with the recommendations and ethical guidelines of the Institutional Animal Care and Use Committee of Yonsei University Wonju College of Medicine (Wonju, Korea). Daily inspections were performed to minimize animal suffering and mice with signs of disease or discomfort were euthanized by CO2 and cervical dislocation. Surgical tissue isolations were performed as terminal procedures under anesthesia.

Please amend your current ethics statement to confirm that your named ethics committee specifically approved this study.

For additional information about PLOS ONE submissions requirements for ethics oversight of animal work, please refer to http://journals.plos.org/plosone/s/submission-guidelines#loc-animal-research  

Once you have amended this/these statement(s) in the Methods section of the manuscript, please add the same text to the “Ethics Statement” field of the submission form (via “Edit Submission”)

Reviewers' comments:

Reviewer's Responses to Questions

**Comments to the Author**

1. Is the manuscript technically sound, and do the data support the conclusions?

Reviewer #1: Partly

Reviewer #2: Partly

2. Has the statistical analysis been performed appropriately and rigorously? 

Reviewer #1: Yes

Reviewer #2: Yes

3. Have the authors made all data underlying the findings in their manuscript fully available?

Reviewer #1: Yes

Reviewer #2: Yes

4. Is the manuscript presented in an intelligible fashion and written in standard English?

Reviewer #1: Yes

Reviewer #2: Yes

5. Review Comments to the Author

Reviewer #1: In the present study "Klotho has protective effects on podocyte injury under diabatic conditions" the authors discussed the protective effect of Klotho on palmitate induced podocyte damage, and the possible ROS elimination related mechanisms. Here are some major concerns:

1. The authors detected the membrane bound Klotho in Figure 1 but discussed the function of recombinant circulating Klotho in the subsequent data. To make a better consistence, the authors should detect the circulating Klotho level in the diabetes patients and mice.

2. Is figure 1B, figure 2C circulating or membrane bound Klotho? If it is membrane bound Klotho, an ELISA on supernatant Klotho should be performed. So should other Klotho expression experiments.

3. The authors made most of the mechanisms related to the anti-ROS function of Klotho. To better confirm the contribution of ROS in podocyte damage, the authors should use an ROS eliminating chemical as a positive control and detect its function on palmitate treated podocytes. Oleate is not a classical or specific ROS eliminating chemical so it cannot be served as a mechanism positive control.

4. Why would Figure 2C no palmitate be higher than control in Klotho expression? Can BSA increase Klotho expression?

5. Full length caspase-3 should not be used to evaluate apoptosis. The authors need to test cleaved caspase-3.

6. The authors made the conclusion to diabetes. However, this research practically works on the effect of palmitate, instead of diabetes itself. The authors should address the conclusion and title as the palmitate effect and make a discussion about the possible involvement in diabetes, as well as the different function of palmitate and oleate in diabetes.

7. The authors need to discuss how ER stress is induced by ROS.

Reviewer #2: In this manuscript entitled ‘Klotho has protective effects on podocyte injury under diabatic conditions’, the authors have attempted to demonstrate the renoprotective effects of Klotho against palmitate induced podocyte injury. The authors claimed that recombinant klotho treatment reduced oxidative stress, inflammatory responses, fibrosis, cytoskeleton changes and cell death in palmitate induced podocyte damage by inhibiting phosphorylation of FOXO3a mediating pathways. This study might suggest substantial protective effects of Klotho against lipid accumulated kidney injury. However, there are several concerns must be addressed.

1) In vitro palmitate treatment was used for diabetic mimic conditions. But it seems insufficient considering pre-existing sustained hyperglycemia in diabetes promotes lipid deposits in diabetic kidney. Please consider include high glucose treatment with or without palmitate groups in the study.

2) For Fig 1, please include IHC of Klotho in diabetic kidney group. Did the expression of Klotho was decreased in glomerular podocytes in diabetic mice model or samples from diabetic patients?

3) Related with previous suggestion from 1 and 2, treatment recombinant Klotho for diabetes mouse models would be recommended to support author’s idea.

4) For in vitro, please indicate which immortalized podocytes were used, mice or human?

5) For Fig 3C, please include cleaved caspase 3 in western blot analysis.

6) For Fig 4A, indicate staining on panel.

7) For Fig 4C, p-elF2a should be normalized to t-elF2a and please check line 271 indicating rKL treatment ameliorates p-elF2a expression. consider include Chop, down stream of Bip-elF2a-ATF4 in ER stress pathway.

8) For Fig 5C, image quality is not good enough to distinguish nuclear translocation of FOXO3a. Also please consider western blot analysis for FOXO3a in cytoplasmic/nuclear fraction.

9) In Line 311, mention regarding MCP1 is not correlate with Fig 6A.

6. PLOS authors have the option to publish the peer review history of their article (what does this mean?). If published, this will include your full peer review and any attached files.

Reviewer #1: No

Reviewer #2: No

---

## [Author Response · Author response to Decision Letter 0]

24 Dec 2020

Dear editor,

We would like to thank you and reviewers of the PLOS ONE for taking the time and effort to review our manuscript. Many of the valuable and constructive comments raised by reviewers were appreciated. Based these comments, we have carefully revised our manuscript as described in the following point-by-point responses. We thank reviewers for their thoughtful comments, which improved our manuscript. 

Reviewer #1

Reviewer #1: In the present study "Klotho has protective effects on podocyte injury under diabatic conditions" the authors discussed the protective effect of Klotho on palmitate induced podocyte damage, and the possible ROS elimination related mechanisms. Here are some major concerns:

1. The authors detected the membrane bound Klotho in Figure 1 but discussed the function of recombinant circulating Klotho in the subsequent data. To make a better consistence, the authors should detect the circulating Klotho level in the diabetes patients and mice.

Answer) We appreciate your comments. In our previous study, we evaluated the association of plasma and/or urine α-klotho with the progression of type 2 diabetic nephropathy (Lee et al., PLoS one, 2014; Kim et al., Journal of Diabetes and Its complication, 2016). The baseline values of plasma and urine α-klotho were measured in 147 patients with type 2 diabetes mellitus, relatively conserved renal function (estimated glomerular filtration rate (eGFR) of ≥60 mL/min/1.73 m2, and serum creatinine < 1.2 mg/dL). In this prospective observational study, a total of 109 type 2 diabetic patients were followed up for 34 months. Patients were divided into tertile groups according to their plasma and urine α-klotho (based on Urine α-klotho/Cr) levels, respectively (Tables 1 and 2). There were no significant differences among the three groups. All patients had a well-conserved renal function, with an eGFR of 93.0 ± 23.2 mL/min/1.73 m2, and the eGFR was not significantly different among the three groups according to plasma α-klotho tertile (p = 0.310). 

Meanwhile, none of the baseline clinical variables differed significantly among the three groups according to baseline urine α-klotho levels (Table 2).

During 34 months of median follow-up period, we found that plasma α-klotho was negatively correlated with the annual decline of eGFR (r = −0.304, P = 0.001; r = 0.042, P = 0.068, respectively), and the development of albuminuria in the early stage of nephropathy of type 2 diabetic patients (eGFR ≥60 mL/min/ 1.73 m2), whereas urinary α-klotho was not (Fig. A, B). Besides, decreased levels of plasma α-klotho predicted the rapid decline of eGFR in a subgroup analysis of normo/microalbuminuric patients (Fig. C).

This study demonstrated that plasma α-klotho was negatively correlated with annual decline in eGFR and the development of albuminuria, which suggests decreased plasma α-klotho might be a predictive marker for the progression of nephropathy with type 2 diabetic patients.

Unfortunately, we were not able to investigate the level of soluble klotho in a diabetic mouse model. Because there was no enough plasma to use for the analysis. However, based on our previous data on patients with type 2 diabetes, we speculate that plasma soluble klotho might be decreased in diabetic mice. We added “plasma soluble klotho is negatively correlated with the progression of nephropathy with type 2 diabetic patients” in the discussion (P18, lines 414~415).

2. Is figure 1B, figure 2C circulating or membrane bound Klotho? If it is membrane bound Klotho, an ELISA on supernatant Klotho should be performed. So should other Klotho expression experiments.

Answer) Figure 1b and Figure 2c show membrane-bound klotho. As suggested, we have performed an ELISA assay to detect soluble klotho in supernatant collected from mouse podocytes treated by BSA control and palmitate. As shown in the figure below, soluble klotho from mouse podocytes was decreased by palmitate treatment compared to control. This data is shown in Fig 2d.

3. The authors made most of the mechanisms related to the anti-ROS function of Klotho. To better confirm the contribution of ROS in podocyte damage, the authors should use an ROS eliminating chemical as a positive control and detect its function on palmitate treated podocytes. Oleate is not a classical or specific ROS eliminating chemical so it cannot be served as a mechanism positive control.

Answer) We agree with your opinion. Monounsaturated fatty acid oleate has a protective effect by restoring palmitate-induced defects, thus we wanted to demonstrate the use of oleate to neutralize palmitate defects in podocytes. As suggested, to confirm the anti-ROS function of klotho in palmitate-induced podocyte damage, we performed DCFDA staining and used ROS scavenger NAC to eliminate ROS as a positive control and observed that palmitate-induced ROS was decreased by rKL like as shown in NAC-treated podocytes. These data are shown in Fig 4a and b.

And rKL restored palmitate-mediated apoptosis-related molecules, Bax and Bcl2, and ER stress molecule Chop at RNA level as shown in NAC-treated podocytes. These data are shown in Fig 3d and Fig 4d.

4. Why would Figure 2C no palmitate be higher than control in Klotho expression? Can BSA increase Klotho expression?

Answer) We appreciate your comments. To investigate the effect of palmitate on klotho expression, we treated podocytes with BSA-conjugated palmitate in a dose-dependent manner (BSA as a CTL, and palmitate at 100, 200, and 400 �M concentration) and carefully rechecked the expression level of klotho. Podocytes that were not treated with palmitate or BSA were marked as 0 �M. As shown in the figure below, BSA-treated control podocytes (CTL) seemed to decrease klotho expression, but not significantly reduced compared to 0 �M. 100 �M of palmitate-treated podocytes seemed to increase expression of klotho, but not significantly. It could be understood as a compensatory or defense reaction to protect against cell injury by palmitate. However, expression of klotho dose-dependently decreased by palmitate treatment. The corrected information and data are shown in Fig 2 legend and Fig 2 c. 

5. Full length caspase-3 should not be used to evaluate apoptosis. The authors need to test cleaved caspase-3.

Answer) We appreciate your comment. As suggested, the expression level of cleaved caspase-3 was investigated using western blot. As shown in the figure below, we observed that the protein levels of cleaved caspase-3 and Bax, known as apoptosis-related proteins were significantly increased in palmitate-treated podocytes as compared to the control cells, whereas the increased expression levels of these proteins were attenuated by rKL. Further, Bcl2 decreased by palmitate treatment was restored by KL. This data indicates that klotho prevents palmitate-induced cytotoxicity in podocytes. The corrected data are shown in Fig 3c.

6. The authors made the conclusion to diabetes. However, this research practically works on the effect of palmitate, instead of diabetes itself. The authors should address the conclusion and title as the palmitate effect and make a discussion about the possible involvement in diabetes, as well as the different function of palmitate and oleate in diabetes.

Answer) We appreciate your comments. As suggested, we changed to write “Protective effects of klotho on palmitate-induced podocyte injury in diabetic nephropathy“ rather than “Klotho has protective effects on podocyte injury under diabetic conditions” in the title (P1, lines 1~2). Palmitate, the most abundant circulating saturated free fatty acid in plasma has been suggested to the major pathogenic mediator in the development of diabetes mellitus and its complication (Weinberg et al., Kidney Int, 2006; Poitout et al., Biochim Biophy Acta, 2010). To discuss the possible involvement of palmitate in diabetes, we added information that “Type 2 diabetic patients showed 1.5- and 3-fold higher plasma palmitate levels compared with healthy subjects (Miles et al., Diabetes, 2003)” in the discussion (P18, lines 407~408). Additionally, we described that “Palmitate is the most abundant circulating saturated FFA in human and rodent plasma (Poitout et al., Biochim Biophys Acta, 2010; Unger et al., Biochim Biophys Acta, 2010) and impairs insulin signaling and increase ER stress, whereas monounsaturated fatty acid oleate has a protective effect by restoring palmitate-induced defects at insulin signaling (Selina et al., Endocr Connect, 2017; P18, lines 404~407).

7. The authors need to discuss how ER stress is induced by ROS

Answer) As suggested, we described that “The accumulation of palmitate induces oxidative stress affecting protein misfolding (Back et al., Annu Rev Biochem, 2012). Excessive production of ROS induced by palmitate participates in ER Ca2+ depletion and ER stress-mediated cell death (Xu et al., Cell Death and Disease, 2015). Our study showed that anti-ROS function of klotho restores the upregulated expression of ER stress-associated proteins and cell death in palmitate-mediated podocyte injury” in the discussion (P19, lines 429~433). 

Reviewer #2

In this manuscript entitled ‘Klotho has protective effects on podocyte injury under diabatic conditions’, the authors have attempted to demonstrate the renoprotective effects of Klotho against palmitate induced podocyte injury. The authors claimed that recombinant klotho treatment reduced oxidative stress, inflammatory responses, fibrosis, cytoskeleton changes and cell death in palmitate induced podocyte damage by inhibiting phosphorylation of FOXO3a mediating pathways. This study might suggest substantial protective effects of Klotho against lipid accumulated kidney injury. However, there are several concerns must be addressed.

1. In vitro palmitate treatment was used for diabetic mimic conditions. But it seems insufficient considering pre-existing sustained hyperglycemia in diabetes promotes lipid deposits in diabetic kidney. Please consider include high glucose (HG) treatment with or without palmitate groups in the study.

Answer) First of all, the authors appreciate your comments. As suggested, we investigated klotho expression in HG- and AGE-treated podocytes using real-time qPCR and found that mRNA level of klotho was reduced by 30mM HG and or 100 �g/ml AGE treatment compared to control. Furthermore, podocytes were treated by NG and (HG+palmitate) with or without rKL and showed decreased protein levels of klotho by (HG+palmitate) treatment, whereas rKL restored decreased klotho expression. These data are shown in Supplementary Figure 1a and b.

2. For Fig 1, please include IHC of Klotho in diabetic kidney group. Did the expression of Klotho was decreased in glomerular podocytes in diabetic mice model or samples from diabetic patients?

Answer) As suggested, we have performed the immunohistochemistry of klotho in normal and diabetic human kidney biopsies. Podocytes in the normal glomerulus showed klotho expression as a cytoplasmic pattern, whereas this expression was decreased in diabetic podocytes. These data are shown in Figure 1a. 

3. Related with previous suggestion from 1 and 2, treatment recombinant Klotho for diabetes mouse models would be recommended to support author’s idea.

Answer) At this point, we do not have the in vivo data demonstrating that recombinant klotho has protective effects on podocyte injury in a diabetic mouse model. To provide the answer to this question, we would like to introduce one reference representing our speculating results. Recently, Oh et al. (Am J Physiol Renal Physiol, 2018) investigated whether klotho exerts a protective effect against glomerular injury in diabetes. To induce diabetic animal models, mice were injected intraperitoneally with streptozotocin and underwent unilateral nephrectomy (UNx). DM/UNx group was treated daily with 10 g·kg-1 ·day-1 of rKL, using an osmotic minipump. They found that a) Klotho attenuates kidney hypertrophy and decreases albuminuria in diabetic mice (Table 1)

b) Klotho inhibits cell cycle arrest and attenuates apoptosis in diabetic kidneys. As show in Figure, increased protein expression of p21, p27, and phospho-p53 in the DM/UNx was significantly attenuated by rKL administration (Figure A). rKL treatment also significantly inhibited the activated ERK1/2 and p38 pathways in the DM/UNx group (Figure B). Furthermore, klotho can reduce cell mortality caused by the diabetic injury. The ratio of protein expression levels Bax/Bcl2 and expression of cleaved caspase-3 were significantly increased in the DM/UNx group, whereas the upregulation of these apoptotic markers was abrogated by rKL (Figure C). TUNEL assay also detected greater numbers of TUNEL positive cells in diabetic glomeruli than in the controls, and rKL administration significantly decreased these numbers (Figure D). 

c) Klotho alleviates diabetic glomerular hypertrophy (Figure A and B) and decreases the increased expression of profibrotic markers, fibronectin (Figure C), and type I collagen (Figure D) in diabetic kidneys. 

4. For in vitro, please indicate which immortalized podocytes were used, mice or human?

Answer) To show the expression of klotho in podocytes as shown in Figure 1b, we used immortalized mouse podocytes and immortalized human podocytes provided by Dr. Peter Mundel and Dr. Moin A. Saleem, respectively. We have used mouse podocytes for other experiments. As suggested, we indicated mouse podocytes in the article.

5. For Fig 3C, please include cleaved caspase 3 in western blot analysis.

Answer) We appreciate your comment. As suggested, the expression level of cleaved caspase-3 was investigated using western blot. As shown in the figure below, we observed that the protein levels of cleaved caspase-3 and Bax, which are known as apoptosis-related proteins, were significantly increased in palmitate-treated podocytes as compared to the control cells, whereas the increased expression levels of these proteins were attenuated by rKL. Further, Bcl2 decreased by palmitate treatment was restored by KL. These data indicate that klotho prevents palmitate-induced cytotoxicity in podocytes. The corrected data are shown in Fig 3c.

6. For Fig 4A, indicate staining on panel.

Answer) We performed DCFDA staining to evaluate the anti-ROS function of klotho in palmitate-induced podocyte damage and added high-resolution images showing clear ROS generation in podocytes as shown in Fig. 4a and b. DAPI was used as a nuclear marker to show the cell numbers.

7. For Fig 4C, p-elF2a should be normalized to t-elF2a and please check line 271 indicating rKL treatment ameliorates p-elF2a expression. consider include Chop, down stream of Bip-elF2a-ATF4 in ER stress pathway.

Answer) We appreciate your comment. As suggested, we described ER stress data. Besides, the ER stress markers, such as Bip, ATF4, and Chop increased in palmitate-treated podocytes, were significantly decreased by rKL as shown in NAC-treated cells. The corrected data are shown in Fig 4c and d.

8. For Fig 5C, image quality is not good enough to distinguish nuclear translocation of FOXO3a. Also please consider western blot analysis for FOXO3a in cytoplasmic/nuclear fraction.

Answer) To provide clear images of FOXO3a expression in palmitate-treated podocytes, we have performed IF staining and quantified total FOXO3a in the nucleus using Image J software. As show below, palmitate-treated podocytes showed decreased levels of FOXO3a in the nucleus compared to BSA control. However, treatment of rKL restored FOXO3a levels in the nucleus compared to palmitate-treated cells. These data are shown in Figure 5c.

9. In Line 311, mention regarding MCP1 is not correlate with Fig 6A.

Answer) We appreciate your comments. The increased protein level of MCP-1 was not clearly shown in palmitate-treated podocytes in Western blot. Thus, we replaced it with MCP-1 mRNA data detected by real-time qPCR. This data is shown in Fig 6a and b.

---

## [Decision Letter · Decision Letter 1]

1 Feb 2021

PONE-D-20-26221R1

Protective effects of klotho on palmitate-induced podocyte injury in diabetic nephropathy

PLOS ONE

Dear Dr. Lee,

Thank you for submitting your manuscript to PLOS ONE. After careful consideration, we feel that it has merit but does not fully meet PLOS ONE’s publication criteria as it currently stands. Therefore, we invite you to submit a revised version of the manuscript that addresses the points raised during the review process.

Your manuscript was reviewed by the same reviewers and one of the reviewers raised few technical points, which require additional round of revision.

We look forward to receiving your revised manuscript.

Kind regards,

Partha Mukhopadhyay, Ph.D.

Academic Editor

PLOS ONE

Reviewers' comments:

Reviewer's Responses to Questions

**Comments to the Author**

1. If the authors have adequately addressed your comments raised in a previous round of review and you feel that this manuscript is now acceptable for publication, you may indicate that here to bypass the “Comments to the Author” section, enter your conflict of interest statement in the “Confidential to Editor” section, and submit your "Accept" recommendation.

Reviewer #1: (No Response)

Reviewer #2: All comments have been addressed

2. Is the manuscript technically sound, and do the data support the conclusions?

Reviewer #1: Yes

Reviewer #2: (No Response)

3. Has the statistical analysis been performed appropriately and rigorously? 

Reviewer #1: Yes

Reviewer #2: (No Response)

4. Have the authors made all data underlying the findings in their manuscript fully available?

Reviewer #1: Yes

Reviewer #2: (No Response)

5. Is the manuscript presented in an intelligible fashion and written in standard English?

Reviewer #1: Yes

Reviewer #2: (No Response)

6. Review Comments to the Author

Reviewer #1: The current study "Protective effects of klotho on palmitate-induced podocyte injury in diabetic nephropathy" made significant improvement compared with the previous version. Here are some more minor concerns to be addressed:

1. Full length of caspase-3 should be shown together with cleaved caspase-3 in western blot

2. TUNEL staining should show nuclear staining in figure 3.

3. The cause-effect relationship of FoxO3a phosphorylation and its nuclear translocation/transcriptional function is not well supported by the reference, which analyzes FoxO1. Phosphorylation and function may be two separate events. The authors should find a correct supporting reference or make a more accurate conclusion.

4. Many groups in the original images of western blot are not statistically analyzed. To reduce confusion, the authors can also analyze these groups, even though they do not contribute much to the conclusion.

5. Judging from Figure 5, the overall expression decrease of FoxO3a is more pronounced in the palmitate group than the phosphorylation increase. The authors should address and explain it.

Reviewer #2: (No Response)

7. PLOS authors have the option to publish the peer review history of their article (what does this mean?). If published, this will include your full peer review and any attached files.

Reviewer #1: No

Reviewer #2: No

---

## [Author Response · Author response to Decision Letter 1]

18 Mar 2021

Dear editor,

We would like to thank you and reviewer #1 for taking the time and effort to review our manuscript. Many of the valuable and constructive comments raised by reviewer #1 were appreciated. Based these comments, we have carefully revised our manuscript as described in the following point-by-point responses. We thank reviewer #1 for his thoughtful comments, which improved our manuscript. 

Reviewer #1 

The current study "Protective effects of klotho on palmitate-induced podocyte injury in diabetic nephropathy" made significant improvement compared with the previous version. Here are some more minor concerns to be addressed:

1. Full length of caspase-3 should be shown together with cleaved caspase-3 in western blot.

Answer) Thank you for your comment. As suggested, we showed cleaved caspase-3 and caspase 3 in Fig 3c.

2. TUNEL staining should show nuclear staining in figure 3.

Answer) Thank you for your comment. For Apo-BrdU TUNEL assays, podocytes were counter stained with propidium iodide (PI) and the Br-dUTP/TdT enzyme. Apoptotic cells showed green fluorescence, whereas red fluorescence was an indication of viable cells. We added this information in the figure legend (P13, lines 294~296). This data is shown in Fig. 3b.

3. The cause-effect relationship of FoxO3a phosphorylation and its nuclear translocation/transcriptional function is not well supported by the reference, which analyzes FoxO1. Phosphorylation and function may be two separate events. The authors should find a correct supporting reference or make a more accurate conclusion. 

Answer) Thank you for your comment. As suggested, we find other supporting references and added the description in the discussion (P20, lines 443~451). “The FOXO transcription factor is an important regulator of longevity and cancer by regulating target genes associated with cellular differentiation, oxidative stress resistance and nutrient shortage [41]. Phosphorylation modification regulates FOXO3a activity through a cytoplasmic-nuclear shuttle mechanism. The PI3K-AKT signaling pathway regulates FOXO3a activity through phosphorylation at three conserved residues (Thr 32, Ser 253, and Ser 315). This phosphorylation excludes FOXO3a from the nucleus and induces binding to the14-3-3 nuclear export proteins, promoting its cytoplasmic accumulation and inhibiting the transactivating activity of FOXO3a [42]. Yamamoto et al. demonstrated that klotho reduces FOXO phosphorylation and promotes its nuclear translocation [4]. Lim et al. showed klotho’s protective role during tacrolimus-induced oxidative stress via FOXO3a-mediated MnSOD expression [43]. In the present study, we found that klotho decreases palmitate-induced FOXO3a phosphorylation resulting in its nuclear translocation and enhances antioxidant expression, including MnSOD. Taken together, this data suggests that klotho protects podocyte dysfunction against palmitate, which might be dependent on the FOXO3a-mediating antioxidant defense system.”

4. Many groups in the original images of western blot are not statistically analyzed. To reduce confusion, the authors can also analyze these groups, even though they do not contribute much to the conclusion.

Answer) Thank you for your comment. As recommended, each group was analyzed and compared in western blots. 

5. Judging from Figure 5, the overall expression decrease of FoxO3a is more pronounced in the palmitate group than the phosphorylation increase. The authors should address and explain it.

Answer) Thank you for your comment. We wanted to investigate that palmitate-induced FOXO3a phosphorylation could be inhibited by recombinant klotho in podocytes. Thus we evaluated expression levels of phosphorylated FOXO3a compared to FOXO3a and found that palmitate increased phosphorylated FOXO3a expression, whereas recombinant klotho reduced it. We changed Figure 5b style to be clearer.

---

## [Decision Letter · Decision Letter 2]

12 Apr 2021

Protective effects of klotho on palmitate-induced podocyte injury in diabetic nephropathy

PONE-D-20-26221R2

Dear Dr. Lee,

We’re pleased to inform you that your manuscript has been judged scientifically suitable for publication and will be formally accepted for publication once it meets all outstanding technical requirements.

Kind regards,

Partha Mukhopadhyay, Ph.D.

Section Editor

PLOS ONE

Additional Editor Comments (optional):

Reviewers' comments:

Reviewer's Responses to Questions

**Comments to the Author**

1. If the authors have adequately addressed your comments raised in a previous round of review and you feel that this manuscript is now acceptable for publication, you may indicate that here to bypass the “Comments to the Author” section, enter your conflict of interest statement in the “Confidential to Editor” section, and submit your "Accept" recommendation.

Reviewer #1: All comments have been addressed

Reviewer #2: All comments have been addressed

2. Is the manuscript technically sound, and do the data support the conclusions?

Reviewer #1: (No Response)

Reviewer #2: (No Response)

3. Has the statistical analysis been performed appropriately and rigorously? 

Reviewer #1: (No Response)

Reviewer #2: (No Response)

4. Have the authors made all data underlying the findings in their manuscript fully available?

Reviewer #1: (No Response)

Reviewer #2: (No Response)

5. Is the manuscript presented in an intelligible fashion and written in standard English?

Reviewer #1: (No Response)

Reviewer #2: (No Response)

6. Review Comments to the Author

Reviewer #1: The current version of "Protective effects of klotho on palmitate-induced podocyte injury in diabetic nephropathy" made a lot of improvement compared with the last version. The authors claimed in the TUNEL stain, PI is to stain for viable cells. However, after fixation, cell membrane is broken and PI stains for all cells despite of viability. The counterstain does not need to distinguish the viable cells from unviable cells. Also, BrdU-TdT enzyme is not part of the counter stain. Please correct the explanation in the figure legend.

Reviewer #2: (No Response)

7. PLOS authors have the option to publish the peer review history of their article (what does this mean?). If published, this will include your full peer review and any attached files.

Reviewer #1: No

Reviewer #2: No

---

## [Editor Report · Acceptance letter]

14 Apr 2021

PONE-D-20-26221R2 

Protective effects of klotho on palmitate-induced podocyte injury in diabetic nephropathy 

Dear Dr. Lee:

I'm pleased to inform you that your manuscript has been deemed suitable for publication in PLOS ONE. Congratulations! Your manuscript is now with our production department. 

Kind regards, 

on behalf of

Dr. Partha Mukhopadhyay 

Section Editor

PLOS ONE